# Effects of Core Antigen Bacterin with an Immunostimulant on Piglet Health and Performance Outcomes When Challenged with Enteric and Respiratory Pathogens

**DOI:** 10.3390/antibiotics12030599

**Published:** 2023-03-16

**Authors:** Charley A. Cull, Vijay K. Singu, Jenna J. Bromm, Kelly F. Lechtenberg, Raghavendra G. Amachawadi, Brooke J. Cull

**Affiliations:** 1Midwest Veterinary Services, Inc., Oakland, NE 68045, USA; vijay@mvsinc.net (V.K.S.); jenna@mvsinc.net (J.J.B.); kelly@mvsinc.net (K.F.L.); brooke@mvsinc.net (B.J.C.); 2Central States Research Centre, Inc., Oakland, NE 68045, USA; 3Department of Clinical Sciences, College of Veterinary Medicine, Kansas State University, Manhattan, KS 66506, USA; agraghav@vet.ksu.edu

**Keywords:** colibacillosis, pastuerellosis, piglets, ETEC, vaccines

## Abstract

A total of 90 pigs, approximately one day of age, were used in a 42-day study to evaluate whether Endovac-Porci, a core antigen vaccine with an immunostimulant, provides piglets with broad-spectrum protection against the enteric and respiratory effects of Gram-negative bacteria. This study was a single-site, randomized, prospective, blinded, comparative placebo-controlled design. Individual pigs were randomly allocated to 1 of 2 treatments in a randomized design. An individual pig was considered the experimental unit for the farrowing phase (Study day 0 to 21), and the pen was considered the experimental unit for the nursery phase (Study day 21 to 42). Thus, there were 45 replications per treatment during the farrowing phase and 15 replications per treatment during the nursery phase. Treatments included a control product (saline; CP) and an investigational product (Endovac-Porci; IVP). On Study day 23, all pigs were challenged with enterotoxigenic *Escherichia coli* strain expressing K88 (F4) fimbriae and *Pasteurella multocida*. Individual pigs were weighed and feed consumption was measured to determine body weight gain, average daily gain, and feed-to-gain ratio. Clinical and fecal scores and overall health were recorded daily. Overall, administering the IVP to pigs led to an increase (*p* < 0.01) in body weight gain and average daily gain compared to pigs administered the CP. Pigs administered the IVP had reduced (*p* < 0.01) mortality compared to pigs administered the CP. There was a Study day × treatment interaction on clinical and fecal scores (*p* < 0.01). There was also a main effect of Study day where clinical and fecal scores increased (*p* < 0.01) as the Study day increased. Treatment also had an effect on clinical and fecal scores, where pigs administered the IVP had lower (*p* < 0.01) clinical and fecal scores compared to pigs administered the CP. In conclusion, administering pigs with the Endovac-Porci vaccination significantly improved the performance (i.e., body weight, body weight gain, and average daily gain) and health (i.e., clinical and fecal scores), while reducing the overall mortality in pigs challenged with *E. coli* K88 orally and *Pasteurella multocida* intranasally post-weaning. Results from this study suggest that Endovac-Porci could provide broad-spectrum protection against enteric and respiratory effects of Gram-negative bacteria in piglets.

## 1. Introduction

Colibacillosis and Pasteurellosis are common bacterial diseases in swine. *Escherichia coli* can affect various ages of pigs differently, causing diarrhea in suckling pigs, neurological issues in nursery pigs, or urinary tract infections in sows. Colibacillosis is an intestinal infection caused by enterotoxigenic *Escherichia coli* (ETEC) that results in significant economic losses [1]. Colibacillosis has a high morbidity and a mortality of up to 70% and is usually more severe in piglets. Strains affecting swine are of low virulence to humans [2]. *E. coli* strains can be differentiated depending on the antigen, such as somatic (O), capsular (K), flagellar (H), and fimbrial (F) genes. Enterotoxigenic *Escherichia coli* (ETEC), which causes neonatal colibacillosis, most commonly carry the fimbriae F4 (K88), F5 (K99), F6 (987P), or F41 gene, while ETEC causing weanling diarrhea frequently carry fimbriae F4 (K88) and F18 genes. These fimbriae adhere to specific receptors on the intestinal epithelial cells (enterocytes), which begin the process of enteric infection. After this, the bacteria produce one or more enterotoxins that induce diarrhea, such as the heat stable toxin a (STa), the heat stable toxin b (STb), and the heat labile toxin (LT) [3,4]. Diagnosing enteric colibacillosis depends on the isolation and quantification of the pathogenic *E. coli* coupled with the demonstration by PCR of the genes that encode for virulence factors (fimbrae and toxins) [1]. Antibiotics are generally used to kill bacteria, and fluids are given to prevent dehydration. Additionally, it is recommended that piglets have a dry and clean environment to prevent infection. Many available vaccines often combine *E. coli* and various species of *Clostridia*. Vaccinating sows two to three weeks before farrowing is a recommended way to increase antibody production and give piglets passive immunity through the colostrum. Some other prevention methods include adding antimicrobials such as carbadox or minerals such as zinc in the first nursery diet [2].

Pneumonic pasteurellosis is often seen along with other diseases that impair respiratory function. There is a primary and secondary form. Primary pasteurellosis usually occurs with meningitis in young pigs. However, this form rarely affects adult swine. Primary pasteurellosis is not as common as secondary pasteurellosis and should only be diagnosed once all other possible causes have been eliminated. Secondary pneumonic pasteurellosis usually follows environmental stresses, which may include dusty or overcrowded conditions, excessive ammonia gas, or poor ventilation. Secondary pneumonic pasteurellosis is often seen alongside *Actinobacillus pleuropneumoniae* and *Mycoplasma pheumoniae*. The etiologic agent is *Pasteurella multocida*, which include toxigenic and non-toxigenic strains. It is isolated frequently from swine herds and inhabits pigs’ nasal passages. Pneumonic pasteurellosis in swine in the United Stated is most often caused by serotypes A and D [4]. So far, there are 16 somatic serotypes and five capsular serotypes (A, B, C, D, E, and F) documented [5,6]. Major economic losses in the pork and poultry industries are expected worldwide. Pneumonic pasteurellosis is very difficult to treat effectively. An early diagnosis, appropriate dosage, and choice of an antimicrobial with sufficient duration is crucial for success. Since pasteurellosis is such a highly contagious disease, affecting almost every animal species, vaccine design strategies that lead to improved, cross-protected vaccines offer the best method of effective control [7,8]. Current control measures can be expensive and have little efficacy, so recently conducted research has been producing more potent and targeted vaccines [9,10]. Recent studies involving the genetic, biochemical, and virulence factors of *P. multocida* and other Pasteurellaceae family members resulted in a greater understanding of the disease mechanisms and the development of new non-bacterin vaccines, many of which are now available in the community for animal use. Several outer membrane proteins on *Pasteurella* have potential targets for vaccine development [7].

To immunologically protect swine from a large spectrum of Gram-negative pathogens, there are many possibilities. A single core antigen common to all Gram-negatives combined with an immunostimulant was tested to determine the delivery of broad-spectrum protection. There are thousands of serotypes in Gram-negative bacteria, which makes it difficult to create effective commercial or autogenous vaccines that protect against a broad-spectrum of Gram-negative pathogens. As a result, it is necessary to find a single bacterin that protects against all Gram-negative pathogens. In the present study, we evaluated whether Endovac-Porci, which is a core antigen vaccine with an immunostimulant, provides piglets broad-spectrum protection against the enteric and respiratory effects of Gram-negative bacteria.

## 2. Results

### 2.1. Performances Responses

There was a main effect of treatment on body weight on Study day 21 and 42. Body weight was higher (*p* < 0.01) in pigs administered the IVP compared to pigs administered the CP (Mean difference = 0.48 kg and 1.5 kg for Study day 21 and 42, respectfully; Table 1).

During the farrowing phase (Study day 0 to 21), there was a main effect of treatment on body weight gain and average daily gain. Pigs administered the IVP had greater (*p* = 0.03) body weight gain and average daily gain compared to pigs administered the CP. A similar result was observed during the nursery phase (Study day 21 to 42), where pigs administered the IVP had greater (*p* < 0.01) body weight gain and an increase (*p* = 0.05) in average daily gain compared to pigs administered the CP (Table 1).

Overall, there was a main effect of treatment on body weight gain and average daily gain. Administering the IVP to pigs led to an increase (*p* < 0.01) in body weight gain and average daily gain compared to pigs administered the CP. There were no observed differences (*p* > 0.05) between treatments for feed-to-gain ratio (Descriptive statistics; Table 1).

### 2.2. Mortality

A total of 29 (32.5%) of pigs died during the entire experimental period. During the farrowing phase (Study day 0 to 21), there was a main effect of treatment on mortality where pigs administered the IVP had reduced (*p* = 0.06) mortality compared to pigs administered the CP. A similar response was observed during the nursery phase (Study day 21 to 42) and the overall experimental period (Study day 0 to 42), where pigs administered the IVP had a reduced (*p* < 0.01) mortality compared to pigs administered the CP (Table 2).

### 2.3. Clinical Scores

Clinical scores were assigned on a 0–3 scale with 3 being severe and 0 being normal (Table 3). There was a Study day × treatment interaction (Table 4; *p* < 0.01). There was also a main effect of Study day where clinical scores increased (*p* < 0.01) as the Study day increased. Treatment also had an effect of clinical score, where pigs administered the IVP had lower (*p* < 0.01) clinical scores compared to pigs administered the CP.

### 2.4. Fecal Scores

Fecal scores, determined at the pen level, were assigned on a 0-3 scale with 3 being severe and 0 being normal (Table 5). There was a Study day × treatment interaction (Table 6; *p* < 0.01). There was a main effect of Study day where fecal scores increased (*p* < 0.01) as the Study day increased. There was also a main effect of treatment on fecal scores. Pigs administered the IVP had lower (*p* < 0.01) fecal scores compared to pigs administered the CP.

## 3. Discussion

Intestinal infection, associated with enterotoxigenic *Escherichia coli* (ETEC), is an important disease that has significant economic impact on the swine industry [1]. Enteropathogenic *E. coli* (EPEC) and enterotoxigenic *E. coli* (ETEC) are the two dominant pathogens that cause enteric colibacillosis. To diagnose enteric colibacillosis, a PCR for virulence factors and quantification and isolation of pathogenic *E. coli* is necessary [1]. ETEC is the most important variety in swine, and it includes several virotypes [1]. Outbreaks of ETEC infection leading to neonatal and post-weaning diarrhea usually affect the same herds and the proper control measures can be expensive [1]. ETEC have fimbriae that elaborate one or more enterotoxins and attach to enterocytes [1]. Additionally, ETEC infections may result in a shock syndrome that causes congestion, hemorrhagic gastroenteritis, renal hemorrhage, and thrombi in the mucosa of the small intestine and the stomach [11]. EPEC is difficult and many veterinary diagnostic laboratories do not screen for this pathotype of *E. coli* routinely [11]. Enterotoxigenic *Escherichia coli* (ETEC) can cause young pigs to lose up to 40% of their body weight and in the worst cases, mortality can reach 100% [12].

Colibacillosis has the potential for human transmission through foodborne illness and has a direct economic impact [12]. The standard treatment is the use of antibiotics, but there is increasing antibiotic resistance, constant consumer pressure, and changing government regulations [12]. Enterotoxigenic *E. coli* produces two main pathogenic determinants, which include fimbria adhesions and toxins [12]. ETEC fimbriae that are most commonly associated with neonatal diarrhea include F4, F5, F6, and F41, while F4 and F18 are more commonly associated with ETEC-induced post-weaning cases diarrhea [12]. Protection from ETEC infection usually is reliant on sow vaccination and passive colostrum antibody immunity [12]. Currently, there are no vaccines available that protect against post-weaning colibacillosis [12].

*Pasteurella* species are commonly found in animal populations and are often found as a part of the normal microbiota or the oral, nasopharyngeal, and upper respiratory tracts [13]. Many *Pasteurella* species are associated with epizootic outbreaks and are opportunistic pathogens [13]. In domestic and wild animals, *Pasteurella* are among the most prevalent opportunistic pathogens [13]. In humans and animals, *Pasteurella* are associated with acute and chronic infections that can lead to significant morbidity and mortality [5]. It is not a common cause of mortality in humans, but deaths caused by pasteurellosis have increased in recent years in the United States [13]. *P. multocida* is often endemic in rabbit colonies and swine herds, where the pneumonia and rhinitis disease are commonly called “sniffles” [13]. Clinical signs of *Pasteurella* include sneezing, deviated snouts, bloody nasal discharge that occur in a large number of grow-finish pigs [2]. Bacteria once colonizes in the lungs leading to bronchopenumonia due to cough and dyspnea in post-weaned pigs [2].

In a recent study, researchers evaluated vaccinated pigs at seven days of age. Pen-group weights were recorded at 28 (weaning), 42 (end of pre-starter phase), and 63 days of life (end of nursery phase) [14]. At each phase, the death culling rates, average daily gain, and average daily feed intake were calculated [14]. Overall, the average daily gain and average daily feed intake were higher in the vaccinated group [14]. Vaccination against edema disease reduced pig losses and improved average daily feed intake and average daily gain [14]. These results are similar to the performance results observed in the current study. Pigs vaccinated with Endovac-Porci had improved performance (body weight gain and average daily gain) and mortality rate compared to pigs that were administered the control (saline). These results, combined with the previously stated findings, suggest that vaccinating pigs to protect against enteric and/or respiratory infection can help improve growth performance and mortality during a Gram-negative bacterial challenge.

One study investigated the effects of *Enterococcus faecium* NCIMB 10415 (*E. faecium*) on intestinal development [15]. Before pigs were challenged with ETEC, there was no significant differences for the average daily gain and fecal scores between the two treatments groups [15]. After the ETEC challenge, the challenged pigs had greater fecal scores compared to the non-challenged pigs. However, pigs administered with *E. faecium* had reduced fecal scores [15]. These results showed that oral administration of *E. faecium* alleviated the intestinal injury and severity of diarrhea of neonatal piglets that were challenged by ETEC by improving intestinal microbiota and immune response [15]. There is more information about F4 ETEC inoculation than F18 ETEC inoculation, and there is high variability in diarrhea response with similar dosages [16]. In another study, two separate ETEC challenge experiments were conducted on the commercial neonate and weanling pigs that received two doses of ESV. These showed a reduction in clinical symptoms and significant reductions in the severity of *E. coli* associated diarrhea [12]. These results are similar to those of the current study, where administering pigs Endovac-Porci can reduce the severity of clinical signs and presence of diarrhea during a Gram-negative, enteric, or respiratory outbreak.

Rectal temperature is a good indicator of pig health as it is one of the best indicators of core body temperature [16]. For an ETEC F4 challenge, the timing can vary from 24 h after inoculation, but it can be very time-consuming and stressful for the animals [16]. Another indicator of infection is bacterial shedding, which differs based on the bacterial species and the timing of the analyses [16]. Assessing and quantifying the pathogenic enterotoxins may be a precise method for controlling the efficacy of the ETEC challenge model since the ETEC toxins indicate the level of infection [16].

One of the main causes of post-weaning diarrhea (PWD) in pigs is enterotoxigenic *Escherichia coli* strains expressing F4 (K88) fimbriae (F4-ETEC) [17]. A study investigated the efficacy of a live oral vaccine that contains a non-pathogenic *E. coli* strain expressing F4 to protect pigs against PWD [17]. Efficacy was evaluated through assessing clinical observations, diarrhea, intestinal fluid accumulation, weight gain, intestinal colonization, and fecal shedding of F4-ETEC [17]. Three days after vaccination, the duration of diarrhea and fecal shedding of F4-ETEC were reduced [17]. Another study determined the prevalence of ETEC in non-diarrheic pigs by sampling 990 pigs from 11 pig farms [18]. There were 19 antibiotic-resistance patterns and 52.5% of them had multiple antibiotic resistances [18]. This study determined that the information generated is important for understanding the ecology and dynamics of ETEC in pigs within various production stages [18].

One study investigated the effects of dietary supplementation of bacteriophages (phages) against enterotoxigenic *Escherichia coli* (ETEC) K88 as a therapy against the ETEC infection in post-weaning pigs [19]. Average daily gain, goblet cell density, and villous height-to-crypt-depth ratio were less in the challenged group than the unchallenged group [19]. This study indicates that the phage therapy is effective for alleviating acute ETEC K88 infection in post-weaning pigs [19]. There is a complex microbial community that influences various aspects of health and development that exist in the gastrointestinal tract of mammals [20]. The outcome of gut colonization in piglets is influenced by the environment, which also emphasizes the developmental window in mammals [20]. One concept in this study is that diseases relevant to humans and growth performance issues relevant to animal production can be linked to the quality of an individual’s intestinal microbiota [20].

In Ontario, post-weaning *Escherichia coli* diarrhea (PWECD) was investigated using a case-control study with 50 nurseries [21]. The hemolytic *E. coli* from 82% of the case herds were positive for 3 enterotoxins, which include STa, STb, and LT) [21]. Additionally, the *E. coli* investigated were resistant to multiple antibiotics [21]. The onset of diarrhea was mostly seen within a week of weaning but was also observed in the grower-finisher stage [21]. PWECD is economically important because mortality was higher and growth rates were worse in case herds [21].

Weaning imposes stress on piglets due to changes in microbiology, immunology, and physiology [22]. The maintenance of a low gastric pH value is essential for a healthy gut because it reduces the passage of pathogenic bacteria into the small intestine [22]. Commensal bacteria in the GIT include the majority of *E. coli*, and they exist in healthy and diseased pigs [23]. Early weaned pigs exhibited a more rapid onset and severity of diarrhea and reductions in weight gain when challenged with ETEC [24]. The early weaning stress can alter immune and physiology responses and clinical outcomes to infectious pathogens [23]. Another study found that an increased risk of PWD was associated with feeding piglets twice a day and feed restriction after weaning [24]. Temperature control decreased the risk of PWD and variation in ambient temperature should be minimized in housing for newly weaned piglets [25]. Pens contaminated with *E. coli* strains are likely the source of infection for piglets, but it could also be passed before weaning [26]. Risk factors for post-weaning diarrhea includes large temperature fluctuations and high creep feed intakes [27].

The present study investigated whether the administration of Endovac-Porci at one day of age and again at the time of weaning improved the performance and health compared to controls after all pigs were challenge with *E. coli* K88 orally and *Pasteurella multocida* intranasally post-weaning. In conclusion, vaccinating pigs with Endovac-Porci significantly improved the performance (i.e., body weight, body weight gain, and average daily gain) and health (i.e., clinical and fecal scores), while reducing the overall mortality in pigs challenged with *E. coli* K88 orally and *Pasteurella multocida* intranasally post-weaning. Results from this study suggest that Endovac-Porcine could provide broad-spectrum protection against enteric and respiratory effects of Gram-negative bacteria in piglets.

## 4. Materials and Methods

All activities related to this study were reviewed and approved by the Institutional Animal Care and Use Committee of Midwest Veterinary Services, Inc. prior to study initiation (IACUC number MVS20006P).

This study was a single-site, randomized, prospective, blinded, comparative placebo-controlled design. This study was conducted at Midwest Veterinary Services, Inc. facility. Pigs were housed appropriate to their stage of life. Nursing pigs were housed in sow crates, and once weaned, pigs were housed in nursery pens with plastic flooring and three pigs per pen. Pigs were housed in a temperature-controlled research facility. Each nursery pen (1.52 × 1.83 m) contained 1 cup waterer and 1 individual self-feeder to allow for *ab libitum* access to feed and water. Pigs in the nursery were allowed approximately 0.927 m^2^ per pig.

### 4.1. Study Population and Animal Management

A total of 90 pigs (Landrace/Duroc cross; initially 1.65 ± 0.08 kg), approximately one day of age, were used in a 42-day study. All pigs were sourced from a commercial farrowing facility (Klitz Farm; West Point, NE, USA). Upon arrival (Study day −1), all pigs were examined prior to randomization by a veterinarian or designee to determine overall health status. Once all pigs on test were deemed healthy and acceptable (i.e., passed an individual physical examination and a clinical and fecal score equal to 0), pigs were processed according to the standard operational procedures of the study site (castrated, tails docked, and administered 1 mL intramuscularly of iron). Pigs were then identified by placing one duplicate numbered ear tag in each ear. Following the farrowing phase (Study days 0 to 21), pigs were weaned on Study day 21 (i.e., 21 days of age), and received a modified live PRRS, PCV2, and *Mycoplasma hyopneumoniae* vaccine prior to being moved into a research nursery facility. During the nursery phase (Study day 21 to 42), pigs were given access to a non-medicated, complete-ration, commercial starter diet that met or exceed the minimum daily nutrient requirements for animals’ age and size. All pigs had daily veterinary oversight as general health observations were performed throughout the study, with standard operational procedures being followed for animal management and care. Sick, injured, or moribund pigs were treated per the standard practice of the study site. Pigs that became moribund, injured, or died were excluded from the study. Moribund pigs were euthanized using an AVMA approved method. In addition, a necropsy and diagnosis were completed for all pigs that died or were euthanized during the study.

### 4.2. Treatment Allocation and Administration

Individual pigs were randomly allocated to 1 of 2 treatment groups in a randomized design using a computer software (Excel, Microsoft Corp., Redmond, WA, USA). An individual pig was considered the experimental unit for the farrowing phase (Study day 0 to 21) and the pen for the nursery phase (Study day 21 to 42). Thus, there were 45 replications per treatment during the farrowing phase and 15 replications per treatment during the nursery phase. Treatment groups were balanced within the litter for the farrowing phase, and on Study day 21, the remaining pigs were randomly allotted to the nursery pen with treatment groups not commingled within a pen during the nursery phase. Treatments included a control product (saline; CP) and an investigational product (Endovac-Porci; IVP). Each pig received 1 mL of the assigned treatment via intramuscular injection. Treatments were administered on Study day 0 (day of birth; administered in the right neck) and Study day 21 (time of weaning; administered in the left neck). All intramuscular injections were performed using a 20 gauge by ½ inch hypodermic needle (Study day 0) or a 18 gauge by 5/8 inch hypodermic needle (Study day 21). A needle was only used for a single pig.

### 4.3. Challenge Pathogens and Procedure

All pigs were challenged with enterotoxigenic *Escherichia coli* strain expressing K88 (F4) fimbriae (O149:LT:Sta:STb:East1:Paa:hemβ:F4) and *Pasteurella multocida* on Study day 23, two days post-booster vaccination and weaning. The *E. coli* was isolated from the feces of a sick 40-day old pig whereas the *P. multocida* was isolated from a lung tissue in a clinically diseased pig. The field isolates were used to mimic diseases commonly encountered when piglets enter a nursery. Both isolates were well characterized by molecular and biochemical tests. The concentration of viable bacteria present in the challenge material was calculated by preparing a serial ten-fold dilutions of the challenge material and plating each dilution on 5% sheep blood agar plates. The plates were incubated for 24 h at 37 °C in a 5% CO_2_ incubator. After incubation, plates inoculated with a sample dilution yielding between 30 and 300 colonies were counted. The colony count was an average of the two plates inoculated with selected dilution. The number of bacteria present in 1 mL of the challenge material was calculated using the following formula:Colonies on Plate × Dilution Factor × 10(1)

Individual pigs were manually restrained and intranasally administered 2 mL (log phase culture containing 1 × 10^9^ CFU/mL ([26]) of *Pasteurella multocida* using an intranasal mucosal atomization device (MAD Nasal^TM^, Morrisville, NC, USA, Teleflex Medical, REF MAD300) attached to a 6 cc luer lock syringe (VetriJec^TM^, Boise, ID, USA, VetOne, Lot no. CLL06044); followed by oral administration of 5 mL of an enterotoxigenic *Escherichia coli* strain expressing K88 (F4) fimbriae [27] (log phase culture containing approximately 1 × 10^10^ CFU/mL) using a 12 Fr feeding tube (Kendall, Covidien^TM^, Boise, ID, USA, REF8890701215 attached to a 6 cc luer lock syringe (VetriJec^TM^, VetOne, Lot no. CLL06044).

### 4.4. Study Outcome Measure and Blinding

All personnel involved in the study were trained on the protocol, the facility standard operation procedures, as study personnel involved in the collection, recording or interpretation of any data was blinded to the treatment assignment of all pigs. The test material dispenser, test material administrator, and quality control personnel with access to the randomization and treatment assignments were unblinded. Unblinded study personnel were not involved in clinical observations (i.e., clinical and fecal scoring, healthy observations, body weight collection, and/or feed weigh in or weigh-backs).

All pigs were individually weighed on Study days 0, 21, and 42. Following weaning, feed was distributed to individual pen feeders as needed. Feed distribution to individual pen and feed weigh back were documented to account for feed consumption. A single feed weighs back occurred on Study day 42. These data were collected to determine body weight gain, average daily gain, and feed-to-gain ratio.

A trained veterinarian evaluated all pigs prior to the enrollment into the study, and from Study day 22 through 35. Daily observations of individual clinical and pen level fecal scores were collected. Clinical and fecal scores were based on categorical scores described in Table 3 and Table 5, respectively.

### 4.5. Statistical Analysis

Performance (body weight gain, average daily gain, and feed-to-gain ratio) parameters and mortality were calculated to determine significant treatment difference after accounting for the design structure and periods (day 0 to 21, 21 to 42, overall, if applicable). Data were analyzed with an individual pig serving as the experimental unit during the farrowing phase and the pen during the nursery phase. Thus, the number of replicates were 45 and 15 per treatment for the farrowing phase and nursery phase, respectively. Data (spreadsheets) were imported into R. Calculated variables (body weight gain, average daily gain [deads-in]) were provided by the investigator and used in models. General and generalized linear mixed models, for continuous and categorical response variables, respectively, were used for analyzes using the lme4 package in R (v 1.1.21). Most models included fixed effects of treatment group and a random effect term for block in order to account for the design structure. The farrowing phase and nursery phase were analyzed separately as pigs within treatment groups were commingled during the farrowing phase and in pens (blocked by weight) in the nursery phase. Models evaluating the overall effect of treatment across both phases (as requested by investigator) were included; however, these models do not adequately address the difference in experimental design across the two phases and should be interpreted with caution. Clinical scores and pen floor scores were analyzed at the pen level using linear mixed models accounting for block and repeated measures on the pens. Differences between treatments were considered significant if *p* ≤ 0.05.

## Figures and Tables

**Table 1 antibiotics-12-00599-t001:** Performance of pigs with model adjusted means by treatment groups used to test core antigen bacterin with an immunostimulant on health and performance outcomes when challenged with enteric and respiratory pathogens.

Item	CP ^1^	SEM ^2^	IVP ^3^	SEM ^2^	*p*
Body weight, kg					
Study day 0	1.61	0.08	1.69	0.08	0.07
Study day 21	6.04	0.44	6.52	0.44	<0.01
Study day 42	11.0	0.70	12.5	0.67	<0.01
Farrowing phase (Study day 0 to 21)					
Body weight gain, kg	4.24	0.43	4.88	0.43	0.03
Average daily gain, kg	0.20	0.02	0.23	0.02	0.03
Nursery phase (Study day 21 to 42)					
Body weight gain, kg	2.61	0.50	4.67	0.47	<0.01
Average daily gain, kg	0.05	0.07	0.25	0.07	0.05
Overall (Study day 0 to 42)					
Body weight gain, kg	6.43	0.58	9.41	0.56	<0.01
Average daily gain, kg	0.17	0.01	0.23	0.01	<0.01
Feed-to-gain ratio, g/kg ^4^	2.27	---	2.19	---	---

^1^ CP: Control Product (saline). ^2^ SEM: Standard Error of the Mean. ^3^ IVP: Investigative Veterinary Product (Endovac-Porci). ^4^ Descriptive statistics are provided for feed-to-gain ratio (*p* > 0.05).

**Table 2 antibiotics-12-00599-t002:** Mortality by treatment groups among pigs used to test core antigen bacterin with an immunostimulant on health and performance outcomes when challenged with enteric and respiratory pathogens.

Treatment Group	Number of Mortalities	Number of Non-Mortalities	*p*
Farrowing phase (Study day 0 to 21)			0.06
CP ^1^	5	40	
IVP ^2^	0	44	
**Treatment Group**	**Mean % (n)**	**SEM ^3^**	** *p* **
Nursery phase (Study day 21 to 42)			<0.01
CP ^1^	40.0 (16)	7.75	
IVP ^2^	18.2 (8)	5.81	
Overall (Study days 0 to 42)			<0.01
CP ^1^	46.7 (21)	7.44	
IVP ^2^	18.2 (8)	5.81	

^1^ CP: Control Product (saline). ^2^ IVP: Investigative Veterinary Product (Endovac-Porci). ^3^ SEM: Standard Error of the Mean.

**Table 3 antibiotics-12-00599-t003:** Clinical score categorization of pigs used to test core antigen bacterin with an immunostimulant on health and performance outcomes when challenged with enteric and respiratory pathogens.

Clinical Score	Description
0	Normal—Alert, active, normal appetite, well-hydrated, coat normal
1	Mild—moves slower than normal, slightly rough coat, may appear lethargic but upon stimulation appears normal, increased respiratory rate
2	Moderate—inactive, may be recumbent but is able to stand, gaunt, may be dehydrated, coughing with increased respiratory rate
3	Severe—down or reluctant, gauntness evident, dehydrated) (will be euthanized)

**Table 4 antibiotics-12-00599-t004:** Clinical scores of pigs with model adjusted means by treatment groups used to test core antigen bacterin with an immunostimulant on health and performance outcomes when challenged with enteric and respiratory pathogens.

Clinical Score ^1^	CP ^2^	SEM ^3^	IVP ^4^	SEM ^3^	Overall ^5^	SEM ^3^
Study day						
22	0.00 ^a^	0.13	0.00 ^a^	0.13	0.00	0.10
23	0.36 ^a^	0.13	0.00 ^b^	0.13	0.18	0.10
24	1.07 ^a^	0.13	0.07 ^b^	0.13	0.57	0.10
25	1.72 ^a^	0.13	0.43 ^b^	0.13	1.07	0.10
26	1.86 ^a^	0.13	0.71 ^b^	0.13	1.29	0.10
27	1.86 ^a^	0.13	0.57 ^b^	0.13	1.22	0.10
28	1.71 ^a^	0.13	0.43 ^b^	0.13	1.07	0.10
29	1.79 ^a^	0.13	0.50 ^b^	0.13	1.15	0.10
30	1.61 ^a^	0.13	0.43 ^b^	0.13	1.02	0.10
31	1.57 ^a^	0.13	0.43 ^b^	0.13	1.00	0.10
32	1.57 ^a^	0.13	0.43 ^b^	0.13	1.00	0.10
33	1.00 ^a^	0.13	0.07 ^b^	0.13	0.54	0.10
34	0.57 ^a^	0.13	0.00 ^b^	0.13	0.29	0.10
35	0.00 ^a^	0.13	0.00 ^a^	0.13	0.00	0.10
Overall ^6^	1.19	0.08	0.29	0.08	---	---

^1^ There was a Study day × treatment interaction (*p* < 0.01). ^2^ CP: Control Product (saline). ^3^ SEM: Standard Error of the Mean. ^4^ IVP: Investigative Veterinary Product (Endovac-Porci). ^5^ Overall average by Study day (*p* < 0.01). ^6^ Overall average by treatment (*p* < 0.01). ^ab^ Means within a row with different superscripts differ (*p* < 0.05).

**Table 5 antibiotics-12-00599-t005:** Fecal score categorization of pigs used to test core antigen bacterin with an immunostimulant on health and performance outcomes when challenged with enteric and respiratory pathogens.

Clinical Score	Description	Blood Present
0	Normal (firm and shaped)—all pigs	Yes or No
1	Soft and shaped	
2	Loose	
3	Watery	

**Table 6 antibiotics-12-00599-t006:** Fecal scores of pigs with model adjusted means by treatment groups used to test core antigen bacterin with an immunostimulant on health and performance outcomes when challenged with enteric and respiratory pathogens.

Fecal Score ^1,2^	CP ^3^	SEM ^4^	IVP ^5^	SEM ^4^	Overall ^6^	SEM ^4^
Study day						
22	0.00 ^a^	0.12	0.00 ^a^	0.12	0.00	0.09
23	1.07 ^a^	0.12	0.36 ^b^	0.12	0.71	0.09
24	2.86 ^a^	0.12	1.93 ^b^	0.12	2.39	0.09
25	2.86 ^a^	0.12	2.50 ^b^	0.12	2.68	0.09
26	3.00 ^a^	0.12	2.14 ^b^	0.12	2.57	0.09
27	2.71 ^a^	0.12	1.71 ^b^	0.12	2.21	0.09
28	2.57 ^a^	0.12	1.29 ^b^	0.12	1.93	0.09
29	2.58 ^a^	0.12	0.86 ^b^	0.12	1.72	0.09
30	2.57 ^a^	0.12	0.86 ^b^	0.12	1.72	0.09
31	2.57 ^a^	0.12	0.57 ^b^	0.12	1.57	0.09
32	2.00 ^a^	0.12	0.29 ^b^	0.12	1.14	0.09
33	1.00 ^a^	0.12	0.00 ^b^	0.12	0.50	0.09
34	0.50 ^a^	0.12	0.00 ^b^	0.12	0.25	0.09
35	0.00 ^a^	0.12	0.00 ^a^	0.12	0.00	0.09
Overall ^7^	1.95	0.07	0.96	0.07	---	---

^1^ Although score data are typically considered as ordinal and not continuous, these data were not robust enough or structured to use non-normal distributions; however, the normal distribution fit these data reasonably well. ^2^ There was a Study day × treatment interaction (*p* < 0.01). ^3^ CP: Control Product (saline). ^4^ SEM: Standard Error of the Mean. ^5^ IVP: Investigative Veterinary Product (Endovac-Porci). ^6^ Overall average by Study day (*p* < 0.01). ^7^ Overall average by treatment (*p* < 0.01). ^ab^ Means within a row with different superscripts differ (*p* < 0.05).

## Data Availability

Not applicable.

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
