# Peer review of "Effects of Core Antigen Bacterin with an Immunostimulant on Piglet Health and Performance Outcomes When Challenged with Enteric and Respiratory Pathogens"

_antibiotics, 2023, doi:10.3390/antibiotics12030599_

Round 1

Reviewer 1 Report

The manuscript entitle "Effects of Core Antigen Bacterin with an Immunostimulant on Piglet Health and Performance Outcomes when Challenged with Enteric and Respiratory Pathogens" has impact in the field of gut health research and development of new vaccine for gram negative bacteria. The Authors evaluated the effect of "Endovac-Porci" on body weight and health performances. They found positive effect body weight and health performances. However, few corrections and quires need to be addressed for the betterment of the manuscript.

Introduction section was well written. However, the mechanism of working of single core antigen bacterin particularly for gram negative bacteria is less focused. In addition, how does the immunostimulant enhance the protection of vaccine is not well focused.

Line no. 46-47: Please add reference.
Line no. 47-55: Please add reference.
Line no. 76-77: Please add reference.
Line no. 80-82: Please add reference.
Line no. 82-84: Please add reference.

Please write note under each table mentioning full form of CP, IVP, SEM.

Graphical presentation of the outcomes is suggested.

Which immunostimulant was used in this study? Is there any composition of the investigational product?
Line no. 326-327: Do the isolates have any accession number for identification? What is source of the isolates? If isolates were identified during this study, then more details of the identification required with reference.
Line no. 339-343: What is the basis of using the mentioned number of E. coli and Pasteurella for infection? Is there any reference?
The authors have stated that the investigating vaccine improved the performance (i.e. body weight, body weight gain, and average 271 daily gain) and health (i.e. clinical and fecal scores), while reducing the overall mortality in pigs. But they did not discuss the mechanism.

The authors scored the feces just based on the appearance. Why have the Authors not compared the fecal pathogen count as an indicator of vaccination effect?

Why have the authors not investigated antibody response against the vaccination?

Why have the Authors not compared the pathological lesions in both groups?

Are there any side-effects of the tested vaccine?

Please add the limitations of the study. In addition, please mention the further research required particularly on the mentioned questions.

Author Response

Reviewer 1

The manuscript entitle "Effects of Core Antigen Bacterin with an Immunostimulant on Piglet Health and Performance Outcomes when Challenged with Enteric and Respiratory Pathogens" has impact in the field of gut health research and development of new vaccine for gram negative bacteria. The Authors evaluated the effect of "Endovac-Porci" on body weight and health performances. They found positive effect body weight and health performances. However, few corrections and quires need to be addressed for the betterment of the manuscript.

Introduction section was well written. However, the mechanism of working of single core antigen bacterin particularly for gram negative bacteria is less focused. In addition, how does the immunostimulant enhance the protection of vaccine is not well focused.

Thank you!

Line no. 46-47: Please add reference.

Agreed and complied.

Line no. 47-55: Please add reference

Agreed and complied.

Line no. 76-77: Please add reference

Agreed and complied.

Line no. 80-82: Please add reference

Agreed and complied.

Line no. 82-84: Please add reference

Agreed and complied.

Please write note under each table mentioning full form of CP, IVP, SEM

We have revised accordingly.

Graphical presentation of the outcomes is suggested.

Thank you. For this kind of clinical trial, we prefer not to use graphical presentation.

Line no. 326-327: Do the isolates have any accession number for identification? What is source of the isolates? If isolates were identified during this study, then more details of the identification required with reference.

We have revised accordingly.

Line no. 339-343: What is the basis of using the mentioned number of E. coli and Pasteurella for infection? Is there any reference?

We have revised accordingly.

The authors have stated that the investigating vaccine improved the performance (i.e. body weight, body weight gain, and average 271 daily gain) and health (i.e. clinical and fecal scores), while reducing the overall mortality in pigs. But they did not discuss the mechanism

We have revised accordingly.

The authors scored the feces just based on the appearance. Why have the Authors not compared the fecal pathogen count as an indicator of vaccination effect?

Authors would like to thank the reviewer for this suggestion. We will consider this approach in our next trial. The present study was not intended to detect or investigate the effect of vaccination on fecal pathogenic bacterial count.

Why have the authors not investigated antibody response against the vaccination?

The present study was not intended to measure the antibody responses. But, we will definitely consider for our future studies.

Why have the Authors not compared the pathological lesions in both groups?

Thank you for the constructive comments and we will definitely consider for our future studies.

Are there any side-effects of the tested vaccine?

During the study period, we did not notice any visible side-effects of the tested vaccine.

Please add the limitations of the study. In addition, please mention the further research required particularly on the mentioned questions.

We have revised accordingly.

Reviewer 2 Report

Dear authors, I very gladly reviewed your article.

I think your research could provide useful information for researchers studying vaccines and concerned with reducing antibiotic use in animal husbandry. In addition, the data presented could help professionals who work alongside and support farmers in making health and management choices.

While the text is fluent and the experimental steps are described in a clear and detailed manner, I report some observations that I hope will help to improve the manuscript:

Abstract: "(Landrace/Duroc cross; initially 1.65 ± 0.08 kg)" unnecessary

Keywords: I would also add "Vaccines"

Introduction: I would expand the literature consulted (there are only three articles that are repeated); add some general information about the use of vaccines in the country where the study was conducted; report more information about the product being tested (Specifically, is it available on the market? In which countries is it used? Are other studies available?)

Results: tables are clear and easy to understand but it would be preferable if they follow the text in the structure

Discussion: it is a bit disjointed in places, I would semplify the initial part (which could be reported in the Introduction paragraph) and focus more attention on the second part, improving the description of the studies cited (e.g. it could be convenient to report the year and country in the text)

Materials and Methods: following bacterial administration, were samples taken to verify infection?

I would write down the conclusion: certainly the data obtained are favorable, but it should be emphasized that it would be important to acquire data from more trials to support the encouraging results

I hope that these suggestions will enhance this manuscript.

Sincerely,
the Reviewer

Author Response

Reviewer 2

Dear authors, I very gladly reviewed your article.

I think your research could provide useful information for researchers studying vaccines and concerned with reducing antibiotic use in animal husbandry. In addition, the data presented could help professionals who work alongside and support farmers in making health and management choices.

While the text is fluent and the experimental steps are described in a clear and detailed manner, I report some observations that I hope will help to improve the manuscript:

Thank you!

Abstract: "(Landrace/Duroc cross; initially 1.65 ± 0.08 kg)" unnecessary

We have revised accordingly.

Keywords: I would also add "Vaccines"

Agreed and complied.

Introduction: I would expand the literature consulted (there are only three articles that are repeated); add some general information about the use of vaccines in the country where the study was conducted; report more information about the product being tested (Specifically, is it available on the market? In which countries is it used? Are other studies available?)

We have revised accordingly.

Results: tables are clear and easy to understand but it would be preferable if they follow the text in the structure

We have revised accordingly.

Discussion: it is a bit disjointed in places, I would semplify the initial part (which could be reported in the Introduction paragraph) and focus more attention on the second part, improving the description of the studies cited (e.g. it could be convenient to report the year and country in the text)

We have revised accordingly.

Materials and Methods: following bacterial administration, were samples taken to verify infection?

We have addressed this comment accordingly.

I would write down the conclusion: certainly the data obtained are favorable, but it should be emphasized that it would be important to acquire data from more trials to support the encouraging results.

Thank you! Yes, we are planning to conduct future clinical trials in the same line.

Round 2

Reviewer 1 Report

Thanks to the Authors for their clarification. Now the manuscript may be published.